# Training/Match External Load Ratios in Professional Soccer Players: A Full-Season Study

**DOI:** 10.3390/ijerph16173057

**Published:** 2019-08-23

**Authors:** Filipe Manuel Clemente, Alireza Rabbani, Daniele Conte, Daniel Castillo, José Afonso, Cain Craig Truman Clark, Pantelis Theodoros Nikolaidis, Thomas Rosemann, Beat Knechtle

**Affiliations:** 1Polytechnic Institute of Viana do Castelo, School of Sport and Leisure, 4960-320 Melgaço, Portugal; 2Instituto de Telecomunicações, Delegação da Covilhã, 6200-001 Covilhã, Portugal; 3Medical and Performance Department, Sporting Clube de Portugal, 2890-529 Lisboa, Portugal; 4Institute of Sport Science and Innovations, Lithuanian Sports University, 44221 Kaunas, Lithuania; 5Faculty of Health Sciences, Universidad Isabel I, 09003 Burgos, Spain; 6Centre for Research, Education, Innovation and Intervention in Sport, Faculty of Sport, University of Porto, 4200-450 Porto, Portugal; 7Centre for Sport, Exercise and Life Sciences, Coventry University, Coventry CV1 5FB, UK; 8Exercise Physiology Laboratory, 18450 Nikaia, Greece; 9Institute of Primary Care, University of Zurich, 8091 Zurich, Switzerland; 10Medbase St. Gallen Am Vadianplatz, 9001 St. Gallen, Switzerland

**Keywords:** association football, training load, load quantification, external load, sports training

## Abstract

The aim of this study was two-fold: (i) to describe the training/match ratios of different external load measures during a full professional soccer season while analyzing the variations between different types of weeks (three, four and five training sessions/week) and (ii) to investigate the relationship between weekly accumulated training loads and the match demands of the same week. Twenty-seven professional soccer players (24.9 ± 3.5 years old) were monitored daily using a 10-Hz global positioning system with a 100-Hz accelerometer. Total distance (TD), running distance (RD), high-speed running (HSR), sprinting distance (SD), player load (PL), number of high accelerations (ACC), and number of high decelerations (DEC) were recorded during training sessions and matches. An individual training/match ratio (TMr) was calculated for each external load measure. Weeks with five training sessions (5dW) presented meaningfully greater TMr than weeks with four (4dW) or three (3dW) training sessions. Additionally, TDratio (TDr) was significantly greater in 5dW than in 3dW (mean differences dif: 1.23 arbitray units A.U.) and 4dW (dif: 0.80 A.U.); HSRr was significantly greater in 5dW than in 3dW (dif: 0.90 A.U.) and 4dW (dif: 0.68 A.U.); and SDr was significantly greater in 5dW than in 3dW (dif: 0.77 A.U.) and 4dW (dif: 0.90 A.U.). Correlations between the weekly training loads and the match demands of the same week were small for PL (r = 0.250 [0.13;0.36]), ACC (r = 0.292 [0.17;0.40]) and DEC (r = 0.236 [0.11;0.35]). This study reveals that ratios of above 1 were observed for specific measures (e.g., HSR, SD). It was also observed that training sessions are not adjusted according to weekly variations in match demands.

## 1. Introduction

Quantification of training/match load represents an important procedure for adjusting training stimuli provided to players for the demands of the match [1,2]. Generally, training load can be classified in [2]: (i) external load, which represents measures derived from position data or inertial measurement units and can be defined as physical demands imposed during training sessions or match scenarios; (ii) internal load, which represents the biological responses of players to a given external load. Training stimuli (external load in particular) should vary in accordance with the type of week (regular, one match per week versus congested, two or more matches per week), the objectives and periodization strategies of the coach, and the types of exercises and their impact of on the players’ dynamics/actions. Thus, training stimuli are highly dependent on the use of specific drills and games during training [3], namely dependent of skill-based drills (e.g., small-sided games and their different formats and task conditions), positional games or full-sized conditioned games. Moreover, when considering match scenarios, there is considerable inter-week variation in terms of determinant external load measures caused by the own dynamics of the match and its contextual factors [4].

Usually, training load can identified with a dose-response relationship consisting in training stimuli and changes in fitness parameters [5,6], which have been extensively used to detect spikes in training load using approaches as the acute:chronic workload ratio. Monitoring training load can be fundamental to reduce the risk of injury, optimizing performance [7] and obtaining an overall idea of how the weekly stimuli differ from the match demands [8]. In the context of soccer, training load has been extensively used to identify injury risk situations (mostly using the acute: chronic workload ratio) [9,10,11,12]. The monitoring cycle is mainly based on the comparison between acute and accumulated load. However, some other relationships can also be analyzed, namely between the load imposed in the week and the match demands that occur in official matches. Despite the importance of understanding the relationships between match demands and accumulated training load (considering the individualization training principle), there is a lack of studies analyzing how such relationships occur in soccer players [4]. The knowledge of the external loads encountered by the players during match-play could be very useful in order to design the training stimuli in each acquisition day of the week [13]. This last concept may particularly improve the individualization of external load based on the match patterns of a given player, as external load may vary significantly in accordance with each player’s role and playing position.

In a study conducted with 12 elite outfield soccer players from an English Premier League team [14], it was observed that in weeks with only one match, there was an accumulated total distance (TD) of approximately 14,000 meters (m) during the training sessions and around 11,000 meters during the match. In the same study [14], it was observed that high-speed running (HSR; distance covered at speeds between 19.8 and 25.1 km·h^−1^) was around 100 meters during training sessions and 900 meters during the match. This suggests that the relationship between accumulated weekly training load and match demands vary in accordance with the nature of the external load measure. Such a fact should be seriously interpreted by sport scientist and strength and conditioning coaches aiming to prepare the players for the worst-case scenario and try to prepare them to extreme events that may increase the risk of injury. In a study conducted on 28 professional Dutch soccer players [8], with the goal of comparing weekly accumulated training load with match demands, it was revealed that weekly training load represents a load equal to 4.4 matches, TD represents the load of 3.1 matches, high accelerations represent the load of 3.9 matches and high decelerations represent the load of 3.3 matches. However, in the same study [8], which considered four training sessions per week, it was observed that running distance and HSR actions are associated with comparably lower relative values, where running distance represents 2.5 matches and high-speed running represents 2.1 matches.

Understanding the ratios between accumulated load and match demands could be useful in identifying the best approach for readjusting stimuli by considering the needs of each player [8]. It is expected that accumulated training load serves as preparation for worst-case scenarios during a match, thus serving as a form of injury risk prevention and an opportunity to develop general fitness levels [12]. Moreover, by considering the relationship between training load and match demands, coaches could organize compensatory training sessions for players with low levels of weekly stimuli while effectively managing players who experience a much more intense load during training when compared to the load they experience during matches [15]. Hence, it seems appropriate to identify external training and match loads with the aim of ensuring appropriate training load distribution during the microcycle.

Even though such relevant information exists, there is a dearth of evidence [8,14] with regard to the relationship between accumulated external training load and match demands. Based on this, the present study aims to: (i) describe the training/match ratios (comparisons between accumulated external load during training sessions and the load occurred in match) during a full professional soccer season while analyzing the variations between different types of weeks; (ii) investigate the relationship (correlations) among weekly accumulated training load and the match demands occurred in the same week. We hypothesize that most demanding speed thresholds will present the lower training/match ratios (mainly considering that most of the training time is used to positional and small-sided games) and that trivial-to-small correlations will be observed between training load and the match demands (considering that is very difficult the match demands of the next match because such demands depend on contextual factors).

## 2. Materials and Methods

### 2.1. Participants

Twenty-seven professional soccer players from the same team (age: 24.9 ± 3.5 years old; stature: 168.8 ± 41.4 cm; body mass: 71.6 ± 18.7 kg; fat mass: 13.6% ± 4.6%) participating in the first Portuguese soccer league were included in this analysis. Of the players, five were external defenders (ED), five were central defenders (CD), four were defensive midfielders (DMF), five were midfielders (MF), five were wingers (WIN) and three were strikers (ST). Goalkeepers were not included in the study based on the limitations imposed by the team’s manager. The playing positions were used as independent variable.

Players were included in the analysis for any given week only if they participated in at least 45 min of the match and participated in all training sessions that week. The players were informed of the study design and the benefits and consequences of their participation. After being informed, they freely signed an informed consent form. The study was approved by a local ethical committee (IPVC-ESDL180711). The design followed the ethical standards of the Declaration of Helsinki for the study in humans.

### 2.2. Study Design

An observational cohort study was implemented on a professional soccer team during a full season (2018/2019). All the training sessions and match demands during the competitive period were monitored using global positioning system (GPS) units. The external load measures monitored were (i) total distance (TD); (ii) running distance (14.0–19.9 km·h^−1^) (RD); (iii) high-speed running (20.0–24.9 km·h^−1^) (HSR); (iv) sprinting distance (> 25.0 km·h^−1^) (SD); (v) player load, which consisted of the estimation of the difference in total acceleration between two consecutive time steps (i.e., the length of the three-dimensional vector of accelerations in the anteroposterior, mediolateral, and craniocaudal axes) (g); (vi) total high accelerations (> 3 m·s^−2^) (ACC); (vii) total high decelerations (> 3 m·s^−2^) (DEC). Walking and jogging distances (< 13.9 km·h^−1^) were excluded based on the low meaning for performance optimization.

Only weeks with one official match and with three or more training sessions were included in this study. This decision was made to reduce the variability among comparisons. Thus, only 22 weeks’ worth of data were included in the analysis. The types of weeks (used as independent variables) were classified based on the number of sessions: (i) weeks with three training sessions (3dW), (ii) weeks with four training sessions (4dW) and (iii) weeks with five training sessions (5dW). Over the course of the study, six weeks were classified as 3dW, 10 weeks were classified as 4dW and six weeks were classified as 5dW.

Ratios of the weekly training and match demands were compared based on the type of week. Moreover, a correlational analysis was performed to analyze the possible dependence between the weekly training load and match demands of specific weeks.

### 2.3. External Load and Training/Match Ratio

Each player’s movements were recorded by a 10-Hz GPS unit (JOHAN Sports, Noordwijk, the Netherlands) during the season. The GPS unit also included an accelerometer, a gyroscope and a magnetometer (100 Hz, 3 axes ± 16 g). A previous study revealed that this GPS model is valid and reliable [16]. The GPS sensor used in the study was tested with a 2.5% ± 0.41% error for total distance covered. Players wore the GPS in a skin-tight bag affixed to the thoracic region between the scapulae. Data collected during the training sessions were stored and analyzed using the JOHAN Sports web application. All the sessions were monitored (from the warm-up to the cooldown). The following measures were recorded daily: (i) TD (meters), (ii) RD (meters), (iii) HSR (meters), (iv) SD (meters), (v) PL (g), (vi) ACC (number) and (vii) DEC (number).

The sum of each measure during all training sessions of the week was calculated per player, thus providing the weekly load for each measure. The match demands (excluding the warm-up) of the players in the same week were also recorded considering the above-mentioned inclusion criteria (namely, the participation of the player in all training sessions and in at least 45 minutes of the match that same week).

A ratio dividing the weekly external load by the match demand of the same week was calculated for the players who competed in the match and who had participated in all training sessions during the week of the match. The training/match ratio (TMr) was calculated for each player (individualized) based on the following formula: TMr = weekly load/match demands.

A TMr was calculated for each external load measure. Therefore, the following TMr values were obtained: (i) TDr (total distance ratio); (ii) RDr (running distance ratio); (iii) HSRr (high-speed running ratio); (iv) SDr (sprinting distance ratio); (v) PLr (player load ratio); (vi) ACCr (accelerations [> 3 m·s^−2^] ratio); and (vii) DECr (decelerations [> 3 m·s^−2^] ratio). Data were collected during what were considered to be good weather and satellite conditions for GPS during training sessions and matches.

### 2.4. Statistical Procedures

The normality and homogeneity levels of the data were tested. The Kolmogorov-Smirnov and Levene tests revealed that all the dependent variables were normally distributed (*p* > 0.05) and homogeneous (*p* > 0.05). To confirm the assumptions of normality and homogeneity in the sample, a two-way ANOVA tested the possible interactions between factors (playing position × type of week) for the dependent external load variables. No significant interactions were found for TDr (*p* = 0.879; partial eta squared = 0.054), RDr (*p* = 0.961; partial eta squared = 0.042), HSRr (*p* = 0.956; partial eta squared = 0.043), SDr (*p* = 0.919; partial eta squared = 0.049), PLr (*p* = 0.768; partial eta squared = 0.065), ACCr (*p* = 0.515; partial eta squared = 0.084) or DECr (*p* = 0.327; partial eta squared = 0.099). For the non-significant interactions, the one-way ANOVA tested the variations in the dependent variables based on the type of week. Tukey’s test was executed to analyze the pairwise comparisons (between types of week) and the standardized differences of Cohen’s d to test the magnitude of changes in the pairwise comparisons. Cohen’s d was calculated and presented using a confidence interval of 95%. Magnitude inferences according to Cohen’s d values were made based on the following thresholds [17]: [0.0–0.2] = trivial; [0.2–0.6] = small; [0.6–1.2] = moderate; [1.2–2.0] = large; and > 2.0, very large. Considering the second objective of the study, a Pearson correlation test was executed to analyze the relationship between the weekly accumulated external load and match values for each investigated variable. The magnitudes of the correlations were interpreted based on the following thresholds: [0.0–0.1] = trivial; [0.1–0.3] = small; [0.3–0.5] = moderate; [0.5–0.7] = large; [0.7–0.9] = very large; [0.9, 1.0] = nearly perfect. Confidence intervals of 95% were used for the correlation values (r). The statistical procedures were executed on SPSS software (version 24.0, IBM, USA) for *p* < 0.05.

## 3. Results

Figure 1 and Figure 2 present the weekly and match external load measures and their TMr values for the three types of week (i.e., 3dW, 4dW and 5dW). The greatest duration, TD, RD and HSR ratios occurred in 5dW. However, it was observed that, in the case of HSR, the ratio varied from 1.1 ± 0.8 to 2.3 ± 1.5 between 3dW and 5dW, thus suggesting that the demand of three days of training is very similar to the demand of one match. PLr, ACCr, and DECr had relatively great values across the different types of weeks, even 3dW. The ratios are relatively high for PLr (2.0 ± 0.6 arbitrary units A.U.), ACCr (2.2 ± 1.8 A.U.), and DEC (1.6 ± 0.9 A.U.).

Standardized differences of TMr between types of weeks are shown in Figure 3. TDr was significantly greater in 5dW compared 3dW (mean differences [dif]: 1.23 A.U.; *p* = 0.000) and 4dW (dif: 0.80 A.U.; *p* = 0.000). RDr was significantly greater in 5dW than in 3dW (dif: 0.84 A.U.; *p* = 0.000) and 4dW (dif: 0.40 A.U.; *p* = 0.005). HSRr was significantly greater in 5dW than in 3dW (dif: 0.90 A.U.; *p* = 0.000) and 4dW (dif: 0.68 A.U.; *p* = 0.000). PLr was significantly greater in 5dW than in 3dW (dif: 1.34 A.U.; *p* = 0.000) and 4dW (dif: 0.85 A.U.; *p* = 0.000). ACCr was significantly greater in 5dW than in 3dW (dif: 1.46 A.U.; *p* = 0.000) and 4dW (dif: 0.84 A.U.; *p* = 0.000). Finally, DECr was significantly greater in 5dW than in 3dW (dif: 1.34 A.U.; *p* = 0.000) and 4dW (dif: 0.77 A.U.; *p* = 0.000).

Cohen’s d represents the difference of A-B. A positive value indicates that A has a greater average than B; a negative value indicates that B has a greater average than A. TDr: accumulated training/match demand ratio of total distance; RDr: accumulated training/match demand ratio of running distance; HSRr: accumulated training/match demand ratio of high-speed running; SDr: accumulated training/match demand ratio of sprinting distance; PLr: accumulated training/match demand ratio of player load; ACCr: accumulated training/match demand ratio of acceleration > 3 m·s^−2^; DECr: accumulated training/match demand ratio of deceleration > 3 m·s^−2^; 3dW: week with three training sessions; 4dW: week with four training sessions; 5dW: week with five training sessions.

Correlations between weekly accumulated external load measures and the demands of the match of the same week were tested. Scatter plots (N = 176) of the measures can be found in Figure 4. Small correlation coefficients were found between weekly load and match demands of the same week for RD (r = 0.156[0,03;0,28]; *p* = 0.038), HSR (r = 0.153[0.03;0.27]; *p* = 0.042), SD (r = 0.125[0.00;0.25]; *p* = 0.098), PL (r = 0.250[0.13;0.36]; *p* = 0.001), ACC (r = 0.292[0.17;0.40]; *p* = 0.001) and DEC (r = 0.236[0.11;0.35]; *p* = 0.002). A trivial correlation was found for TD (r = 0.030[−0.09;0.15]; *p* = 0.691).

## 4. Discussion

The results of the present study reveal that the training/match ratios tend to vary from ~2 to 4 A.U. considering the external load measure (with the exceptions of the ratios for high-speed running and sprinting distance, which were relatively low). However, such ratios are dependent on the number of training sessions per week. The investigated comparisons revealed that weeks with five training sessions had statistically greater workload ratios than weeks with three or four training sessions. Moreover, it was observed that an increase in the number of training days leading to higher training/match ratio. Finally, trivial-to-small correlation coefficients were observed between the weekly training load and the match demands of that week.

A previous study conducted on Dutch professional players [8] was the first, to the author’s knowledge, to report on the relationship between weekly training load and match demands. Our study entailed a similar approach by using ratios obtained by dividing the sum of the weekly external load by the match demands of that week for each player. Our results revealed that total distance, player load and total number of high (> 3 m·s^−2^) accelerations and decelerations in training sessions were much greater in comparison to the match during the same week, with values of 1.8 ± 0.6, 2.0 ± 0.6, 2.2 ± 1.8 and 1.6 ± 0.9, respectively, in weeks with only three training sessions; these values reached 3.5 ± 1.3, 3.8 ± 1.6, 4.1 ± 1.6 and 3.4 ± 1.9, respectively, in weeks with five training sessions. The training/match ratios for running distance and high-speed running distance were 1.2 ± 0.7 and 1.1 ± 0.8, respectively, in 3dW and 2.3 ± 1.3 and 2.3 ± 1.5, respectively, in 5dW. This suggests that the number of training sessions tend to emphasize the stimuli of overall distance and accelerations/decelerations.

A previous study conducted on professional soccer players from the English Premier League [18] revealed that in weeks with four training sessions, the total distance covered was 19,939 meters, the high-speed running distance covered was 398 meters, the sprinting distance covered was 87 meters and player load reached 2093 A.U. Our results for position data are somewhat similar, since in 4dW, we recorded 20,367 meters of total distance, 442.5 meters of high-speed running and 80.9 meters of sprinting distance. However, the value we obtained for player load was lower (1139.5 g) than those reported in the study on Premier League players [18]. Moreover, our results differ from the descriptive statistics presented in another study conducted in professional soccer players from the Premier League, where players completed 880 meters of running distance, 156 meters of high-speed running and 8 meters of sprinting distance during 4dW [14]. The context and the coach’s decisions for designing drills may have contributed to these differences.

When considering the ratios of weekly training load to the reported match demands of that week, we found quite different values compared with a previous investigation in English Premier League players [14], in which ratios of 0.56 for running distance, 0.22 for high-speed running and 0.03 for sprinting were registered. These results are much different from our results and those reported in Dutch players [8]. A possible explanation for these differences might be that, in the current study, the conditioning training was generally included within the training sessions, whereas in the English Premier League [14], players might have performed conditioning exercises in supplementary sessions. This, however, is only speculation, as this is not reported in the article, and therefore highlights a viable avenue for future research (taken into account when technical staff quantify the training sessions).

Conversely, our results seems in line with those documented in Dutch players [8] in which ratios for 4dW were 3.1 for total distance (versus a ratio of 2.8 was recorded in our study), 2.5 for running distance (versus a ratio of 2.0 was recorded in our study), 2.1 for high-speed running (versus a ratio of 2.0 was recorded in our study), 3.9 for high accelerations (versus a ratio of 3.5 was recorded in our study) and 3.3 for high decelerations (versus a ratio of 2.8 was recorded in our study). Therefore, our study and the study conducted on Dutch players [8] provides clear evidence that accelerations/decelerations have the greatest training/match ratios of all variables observed. This may be partially explained by the typology of drills used, where, usually, in the Portuguese training coaching philosophy, small-sided games are preferred. These games increase the frequency of accelerations/decelerations while decreasing opportunities to perform high-speed running or sprinting [19,20]. Therefore, the use of small-sided games may be part of an explanation to justify the increase of acceleration/deceleration load when compared to the match. For that reason, coaches should consider to properly control the stimuli and manage players during the week.

The second purpose of this study was to investigate the relationship between accumulated weekly training loads and the weekly match demands. Our results revealed trivial-to-small correlations between the weekly accumulated loads and match demands. This suggests that training is independent of the dynamics of the next match. This is due to the nature of the match (i.e., the fact that matches involve uncertainty based on contextual factors) and possibly also due to variations in the time available to train for the next match (i.e., different number of sessions). Week-by-week training may be the most relevant association that can be made to understand why the dynamics of the weekly accumulated load and the match are not highly correlated. As far as we know, no studies have tested this hypothesis calling for further investigations.

Although this study indicates valuable practical implications for football coaches and practitioners, it presents some limitations. The small sample size and the fact that only one team was analyzed can be both considered as a limitation. However, these are very common obstacles in studies involving professional and elite players. Another limitation is that we included players only if they participated in all training sessions during a week and that week’s match, thus reducing the sample size. Nevertheless, this choice was made to guarantee that precise training/match ratios were obtained without considering the average values throughout the season (ratios were calculated weekly). Finally, we did not consider any internal load measures (e.g., heart rate). However, this was out of the scope of our study. Despite of the study limitations, this study clearly reveals that there is a considerable variation in the external load training ratios, and this may be carefully interpreted based on the fact that a different training profile have been applied comparing to the real stimulus of the match. Moreover, some determinant external load measures (e.g., HSR or sprinting) are clearly undertrained comparing with more prevalent measures (e.g., TD, ACC or DEC) and this is also an important evidence to highlight in this study.

Future studies could improve the overall design of the present study by analyzing intra-week variations and identifying which days or types of drills most significantly contribute to increases or decreases in certain measures. This would make it easier to determine how to manage training/match ratios. Future studies should investigate which types of ratios (threshold values) improve performance and reduce the risk of injury. This seems to be the most appropriate manner to understand which type of TMr is recommended and, maybe, an individualized effect will occur in which some players will need greater or lower TMr than others to improve the performance or to decrease the injury risk. Moreover, progressions in TMr should be also carefully researched aiming to understand the beneficial undulations across the weeks.

This study has several practical implications. We would like to emphasize that this study was only the second study (as far as we know) to compare weekly training demands with match demands. The findings presented here constitute important information for coaches. In addition to proposing a simple method for comparing weekly training load considering the individualization of the players (i.e., training/match ratio), we demonstrated that the accumulated total distance and number of high accelerations/decelerations during the training sessions are three to four times greater than the demands of the average match. Demands for running distance, high-speed running distance and sprinting distance were one to two times the demands of a match. Moreover, we revealed that increases in the number of training sessions per week greatly contribute to increases in certain training/match ratios. Based on this evidence, coaches may consider readjusting the training stimulus over a week by allowing more time to stabilize (i.e., make more regular) training/match ratios across the week. This, however, may interfere with other variables (e.g., acute:chronic workload ratio) and, therefore, should be carefully managed to optimize the players’ performance and reduce the risk of injury. Finally, using the results found in our study it is possible to observe that to achieve a TMr of 2.0 (as an example) only 3 sessions/week are necessary, but in the case of HSR, 4 sessions/week are necessary. This certainly will affect some decisions made during the planning of sessions.

## 5. Conclusions

The first purpose of the present study was to describe training/match ratios in professional soccer players and to compare the ratios among weeks with different numbers of training sessions. Results revealed that weeks with five training sessions had statistically greater values for all external load ratios than weeks with three or four sessions. Moreover, it was observed that specific variables (e.g., high-speed running distance and sprinting distance) were associated with substantially lower ratios than other variables. It might be necessary for coaches to schedule more training sessions to promote differences between weekly accumulated training load and the load imposed in a single match.

The second purpose of this study was to determine the relationship between the accumulated load of each week with the weekly match demand. Results revealed trivial-to-small correlations depending on the type of external load analyzed. However, since matches are dynamic and unpredictable in some respects, it may be impossible for accumulated weekly loads and their variations to be adjusted according to match loads.

## Figures and Tables

**Figure 1 ijerph-16-03057-f001:**
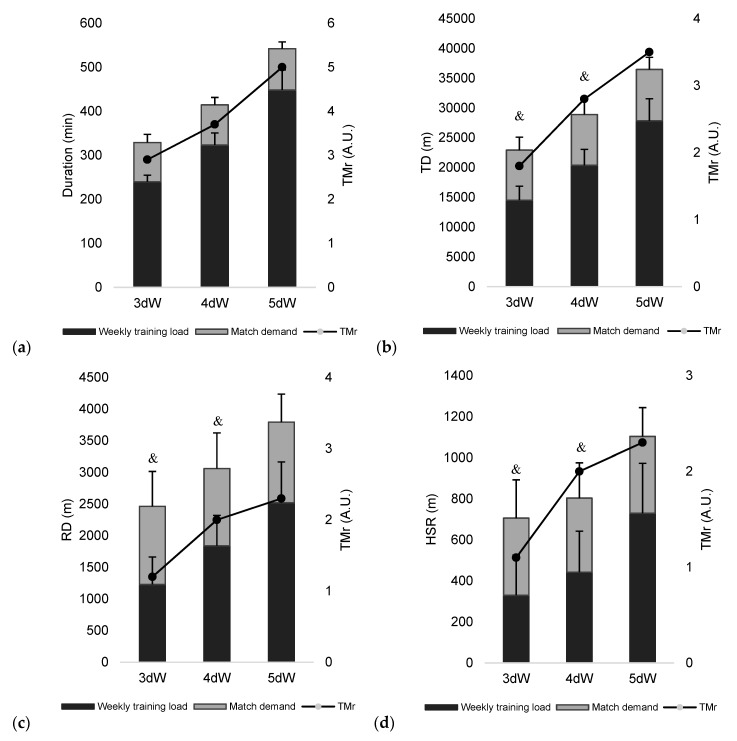
Accumulative weekly (**a**) duration; (**b**) total distance (TD); (**c**) running distance (RD); (**d**) high-speed running (HSR) and their training/match ratios (TMr). &: significantly different from 5dW.

**Figure 2 ijerph-16-03057-f002:**
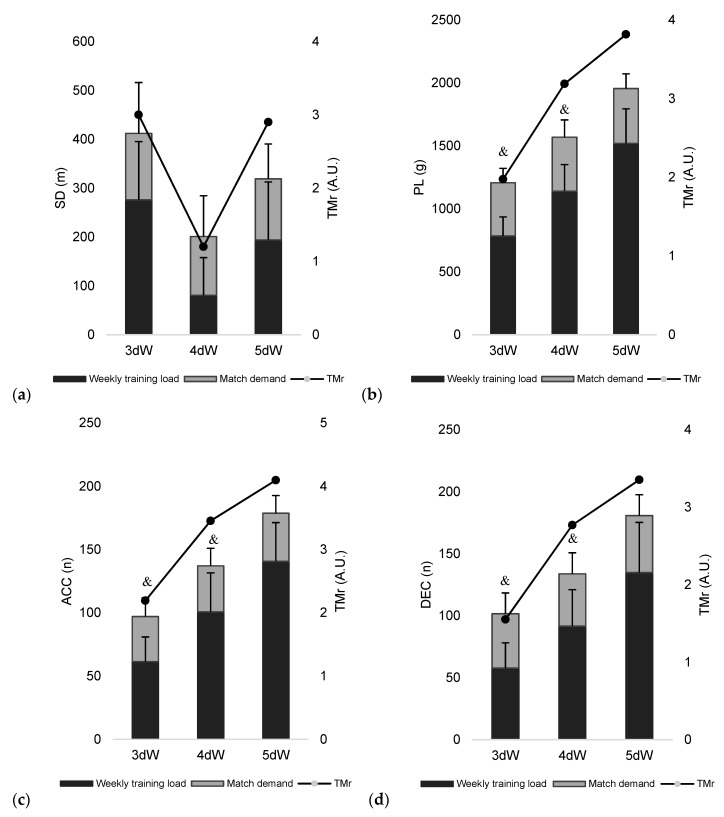
Accumulative weekly (**a**) sprinting distance (SD); (**b**) player load (PL); (**c**) number of accelerations (ACC); (**d**) number of decelerations (DEC) and their training/match ratios (TMr). &: significant different from 5dW.

**Figure 3 ijerph-16-03057-f003:**
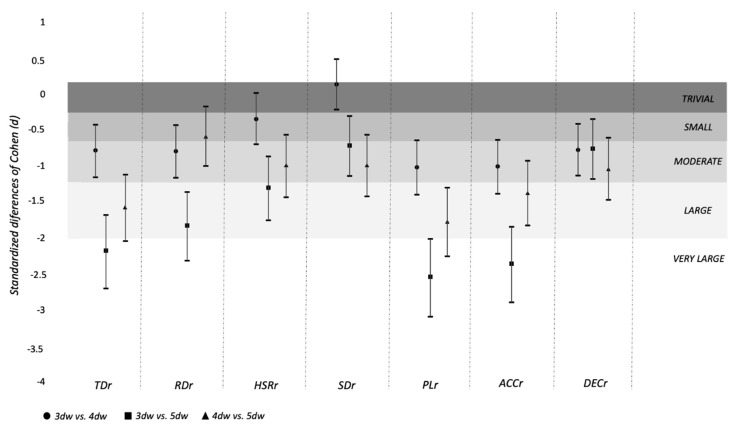
Standardized differences of the different external load ratios between types of week (overall data, not split by playing positions).

**Figure 4 ijerph-16-03057-f004:**
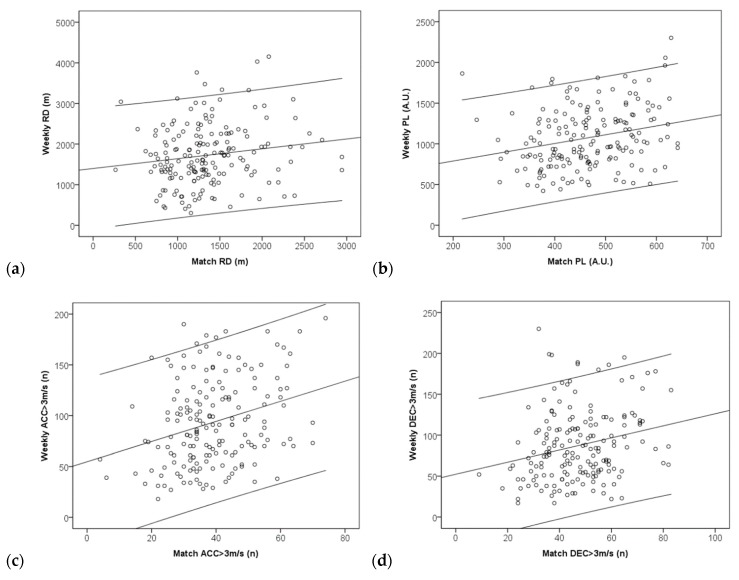
Scatter plots of relationships between accumulated weekly training external load and the match demands for the same week: (**a**) running distance, (**b**) player load, (**c**) total accelerations (>3 m·s^−2^); (**d**) total decelerations (>3 m·s^−2^).

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
