# Peer review of "Training/Match External Load Ratios in Professional Soccer Players: A Full-Season Study"

_ijerph, 2019, doi:10.3390/ijerph16173057_

Round 1
Reviewer 1 Report
1. Pg. 2 line 63-64 few studies have described and characterized the relationships between accumulated training load and real match demands.
The meaning of this sentence is difficult to understand. What do you want to say? What relation between accumulated training load and real match demands? Please write down concretely.
2. What is your hypothesis in this study? In the introduction, you should mention the hypothesis clearly, particularly, how the relationship between accumulated training load (demands?) during training session and match demands (load?) change in accordance with changes in the training session per week or the number of match per week?
3. line 99. (i) describe the different external load measures of the training/match ratios? The sentence is difficult to understand.
4. line 100. between → among different number of training session per week?
5. line 102. What kind of the relationship?
Even if you understand, others can not understand the meaning.
6. In figure 1 & 2, please put the mark of the statistically significant level.
7. In figure 4, please write down the statistically significant level (correlation coefficient, p value and the number of plots).
8. In the result, you found that if the number of training session increase(3 to 5 day/wk), then workload increase and overall running distance, accelerations, decelerations also increase. These findings is very natural. From these results, what is suggested?
9. line 272. that that?
10. In this study, it was revealed the duration, TD, RD, HSR etc. of accumulated training session matches. However, what does these increase mean physiologically and biologically? As you did not express objective parameter, the readers can not understand the meaning of result. The results of this study is very natural and anybody can image. You should mention of the meaning of training load's increase. Is this good or bad? And why?
If you measured, for example, heart rate during training session and matches, you can describe HR during training session and matches and the load (stimuli) to body based on HR data.
Author Response
REVIEWER 1
Pg. 2 line 63-64 few studies have described and characterized the relationships between accumulated training load and real match demands.
The meaning of this sentence is difficult to understand. What do you want to say? What relation between accumulated training load and real match demands? Please write down concretely.
AUTHORS: DEAR REVIEWER, THANK YOU. WE HAVE CHANGED AIMING TO CLARIFY THE IDEA.
What is your hypothesis in this study? In the introduction, you should mention the hypothesis clearly, particularly, how the relationship between accumulated training load (demands?) during training session and match demands (load?) change in accordance with changes in the training session per week or the number of match per week?
AUTHORS: DEAR REVIEWER, THANK YOU. WE HAVE ADDD THE STUDY HYPOTHESIS IN THE END OF THE LAST PARAGRAPH (OF INTRODUCTION SECTION).
line 99. (i) describe the different external load measures of the training/match ratios? The sentence is difficult to understand.
AUTHORS: DEAR REVIEWER, THANK YOU. WE HAVE CHANGED AIMING TO CLARIFY THE SENTENCE.
line 100. between → among different number of training session per week?
AUTHORS: DEAR REVIEWER, THANK YOU. WE HAVE CHANGED ACCORDINGLY.
line 102. What kind of the relationship?
Even if you understand, others can not understand the meaning.
AUTHORS: DEAR REVIEWER, THANK YOU. WE HAVE ADDED THE CONCEPT OF CORRELATION.
In figure 1 & 2, please put the mark of the statistically significant level.
AUTHORS: DEAR REVIEWER, THANK YOU. WE HAVE ADDED SYMBOLS IN THE FIGURES.
In figure 4, please write down the statistically significant level (correlation coefficient, p value and the number of plots).
AUTHORS: DEAR REVIEWER, THANK YOU. WE HAVE ADDED SUCH INFORMATION (P AND R) IN THE RESULTS DESCRIPTION. THE INFORMATION ABOUT N OF PLOTS WAS ALSO ADDED IN THE RESULTS.
In the result, you found that if the number of training session increase (3 to 5 day/wk), then workload increase and overall running distance, accelerations, decelerations also increase. These findings is very natural. From these results, what is suggested?
AUTHORS: DEAR REVIEWER, THANK YOU. IN FACT, OUR AIM WAS TO DESCRIBE SUCH EVIDENCE AND COACHES OR SPORT SCIENTISTS MAY USE THE INFORMATION TO IDENTIFY HOW MANY TRAINING SESSIONS ARE NECESSARY TO ACHIEVE A GIVEN TMr. EXAMPLE, TO ACHIEVE A TMr OF 2.0 IN TOTAL DISTANCE IT IS ONLY NECESSARY 3 SESSIONS/WEEK BUT FOR HSR IS NECESSARY 4 TRAINING/WEEK. WE HAVE USED THIS EXAMPLE TO EXTEND THE DISCUSSION OF PRACTICAL APPLICATIONS.
line 272. that that?
AUTHORS: DEAR REVIEWER, THANK YOU. WE HAVE REMOVED ONE OF THE “THAT”.
In this study, it was revealed the duration, TD, RD, HSR etc. of accumulated training session matches. However, what does these increase mean physiologically and biologically? As you did not express objective parameter, the readers can not understand the meaning of result. The results of this study is very natural and anybody can image. You should mention of the meaning of training load's increase. Is this good or bad? And why?
If you measured, for example, heart rate during training session and matches, you can describe HR during training session and matches and the load (stimuli) to body based on HR data.
AUTHORS: DEAR REVIEWER, THANK YOU. ACTUALLY, WE DO AGREE THAT OUR STUDY HAS SOME LIMITATIONS BASED ON THE FACT THAT FURTHER INVESTIGATIONS SHOULD CONSIDER TO ANALYZE THE the most appropriate manner to understand which type of TMr are recommended and, maybe, an individualized effect will occur in which some players will need greater or lower TMr than others to improve the performance or to decrease the injury risk. Moreover, progressions in TMr should be also carefully researched aiming to understand the beneficial undulations across the weeks. WE HAVE ADDED SUCH FACT IN THE SUB-SECTION OF STUDY LIMITATIONS AND FUTURE STUDIES.
Reviewer 2 Report
I would like to thank the authors for the study conducted. The results found will be very interesting for society. The objectives are clear and the manuscript presents a great coherence from the abstract to the references. The introduction places the readers in a suitable starting point for the full understanding of the article. Therefore, I believe that the manuscript can be published. However, I would like to suggest some minor changes that could improve the quality of the article:
In the abstract, line 32, it says "weeks with...". How many weeks?
Why goalkeepers were not included in the study? Justify
Why did you choose these external loads? Justify
In the conclusion section, the results are repeated again, and some suggestions are done. Nevertheless, I would suggest writing conclusions instead of recommendations for coaches. These recommendations can be included in the last paragraph of the discussion section.
Congratulations to the authors. I hope to see the article published soon.
Author Response
REVIEWER 2
I would like to thank the authors for the study conducted. The results found will be very interesting for society. The objectives are clear and the manuscript presents a great coherence from the abstract to the references. The introduction places the readers in a suitable starting point for the full understanding of the article. Therefore, I believe that the manuscript can be published.
AUTHORS: DEAR REVIEWER, THANK YOU SO MUCH FOR YOUR COMMENTS AND SUGGESTIONS.
However, I would like to suggest some minor changes that could improve the quality of the article:
In the abstract, line 32, it says "weeks with...". How many weeks?
AUTHORS: DEAR REVIEWER, THANK YOU. WE HAVE ADDED SUCH INFORMATION IN THE ABSTRACT.
Why goalkeepers were not included in the study? Justify
AUTHORS: DEAR REVIEWER, THANK YOU. WE HAVE ADDED A JUSTIFICATION IN-TEXT (SUB-SECTION OF PARTICIPANTS)
Why did you choose these external loads? Justify
AUTHORS: DEAR REVIEWER, THANK YOU. WE HAVE ADDED A JUSTIFICATION IN-TEXT (SUB-SECTION OF STUDY DESIGN)
In the conclusion section, the results are repeated again, and some suggestions are done. Nevertheless, I would suggest writing conclusions instead of recommendations for coaches. These recommendations can be included in the last paragraph of the discussion section.
AUTHORS: DEAR REVIEWER, THANK YOU. WE HAVE REMOVED THE PRACTICAL IMPLICATIONS FROM THE CONCLUSIONS.
Congratulations to the authors. I hope to see the article published soon.
AUTHORS: DEAR REVIEWER, THANK YOU SO MUCH FOR YOUR COMMENTS AND SUGGESTIONS.
Reviewer 3 Report
This is an interesting study that seeks to exam training load and match load across a soccer season. This should be of interest to practitioners and coaches who are involved in maintaining performance across a season.
Comments:
Introduction -p2 line 53 expand the discussion of training drills and training stimuli eg use of small sided games to mimec game situations.
p2 line 66 remove the term very interesting as this is personal opinion
p2 line 75 the sentence that starts on this line is too long and should be broken up or shortened.
p2 line 87 there are no references in this section which means you are just giving your personal opinion. Either find evidence to support what you are saying or remove this paragraph.
Methods - p3 line 126 need to justify your choice of metrics.
p4 line 149 - 164 this seems like repetition of what has been said previously. I would suggest that you rewrite the methods section to remove any repetition.
Discussion - there has been an attempt to compare the findings to previous literature but I find that the discussion does not move beyond this.
p9 - this whole page is just personal opinion. The discussion needs to be rewritten to explore what the data means for soccer performance or injury risk and how this data should be used by practiotioners.
Round 2
Reviewer 1 Report
I confirmed your modification. The quality improved.
Please revise English (particular grammar).
Reviewer 3 Report
The authors have addressed major concerns